# Targeted detection and quantitation of histone modifications from 1,000 cells

**Nebiyu A. Abshiru[1], Jacek W. Sikora[1], Jeannie M. Camarillo[1], Juliette A. Morris[1], Philip D. Compton[1], Tak Lee[2], Yaseswini Neelamraju[3], Samuel Haddox[3], Caroline Sheridan[2], Martin Carroll[4], Larry D. Cripe[5], Martin S. Tallman[6], Elisabeth M. Paietta[7], Ari M. Melnick[2], Paul M. Thomas[1], Francine E. Garrett-Bakelman[2,3,8] \*, Neil L. Kelleher[1] \***

**1** Departments of Chemistry, Molecular Biosciences, and the Proteomics Center of Excellence, Northwestern University, Evanston, IL, United States of America, **2** Division of Hematology/Oncology, Department of Medicine, Weill Cornell Medicine, New York, NY, United States of America, **3** Department of Biochemistry and Molecular Genetics, University of Virginia, Charlottesville, VA, United States of America, **4** Division of Hematology and Oncology, University of Pennsylvania Perelman School of Medicine, Philadelphia, PA, United States of America, **5** Indiana University/Melvin and Bren Simon Cancer Center, Indianapolis, IN, United States of America, **6** Memorial Sloan Kettering Cancer Center, New York, NY, United States of America, **7** Montefiore Medical Center—Moses Campus, Bronx, NY, United States of America, **8** Division of Hematology/Medical Oncology, Department of Medicine, University of Virginia, Charlottesville, VA, United States of America

\* fg5q@virginia.edu (FEGB); n-kelleher@northwestern.edu (NLK)

**Data Availability Statement:** All relevant data are within the paper and its Supporting Information files.

## Abstract

Histone post-translational modifications (PTMs) create a powerful regulatory mechanism for maintaining chromosomal integrity in cells. Histone acetylation and methylation, the most widely studied histone PTMs, act in concert with chromatin-associated proteins to control access to genetic information during transcription. Alterations in cellular histone PTMs have been linked to disease states and have crucial biomarker and therapeutic potential. Traditional bottom-up mass spectrometry of histones requires large numbers of cells, typically one million or more. However, for some cell subtype-specific studies, it is difficult or impossible to obtain such large numbers of cells and quantification of rare histone PTMs is often unachievable. An established targeted LC-MS/MS method was used to quantify the abundance of histone PTMs from cell lines and primary human specimens. Sample preparation was modified by omitting nuclear isolation and reducing the rounds of histone derivatization to improve detection of histone peptides down to 1,000 cells. In the current study, we developed and validated a quantitative LC-MS/MS approach tailored for a targeted histone assay of 75 histone peptides with as few as 10,000 cells. Furthermore, we were able to detect and quantify 61 histone peptides from just 1,000 primary human stem cells. Detection of 37 histone peptides was possible from 1,000 acute myeloid leukemia patient cells. We anticipate that this revised method can be used in many applications where achieving large cell numbers is challenging, including rare human cell populations.

## Introduction

Histone post-translational modifications (PTMs) are essential for epigenetic regulation and maintenance of major DNA metabolic processes such as replication, transcription, and

**Funding:** This work was carried out with a financial support from The Paul G. Allen Family Foundation (Award # 11715), NCI CCSG P30CA060553 awarded to the Robert H. Lurie Comprehensive Cancer Center, and the National Resource for Translational and Developmental Proteomics supported by P41GM108569 to NLK. We would also like to acknowledge funding from NCI K08CA169055, UVA Cancer Center through the NCI Cancer Center Support Grant P30CA44579, the University of Virginia and the American Society of Hematology (ASHAMFDP-20121) under the ASH-AMFDP partnership with The Robert Wood Johnson Foundation to FEG-B. This study was conducted by the ECOG-ACRIN Cancer Research Group (Peter J. O'Dwyer, MD and Mitchell D. Schnall, MD, PhD, Group Co-Chairs) and supported by the National Cancer Institute of the National Institutes of Health under the following award numbers: CA180827, CA180820, CA233290, CA189859, CA233321, CA233270. The funders had no role in study design, data collection and analysis, decision to publish, or preparation of the manuscript.

**Competing interests:** The authors have declared that no competing interests exist.

chromatin remodeling [1,2]. The most common histone PTMs include acetylation, methylation, ubiquitination and phosphorylation [3]. These PTMs control gene expression by directly modifying the chromatin structure and/or by providing a docking site for the recruitment of other chromatin modifying complexes [4]. For instance, acetylation of histone H4 K16 promotes gene expression by inhibiting chromatin condensation and formation of higher order chromatin fibers [5]. In contrast, trimethylation of histone H3 K9 is widely known to promote gene silencing via chromatin compaction [6,7]. Disruption of basal levels of histone modifications can lead to genome instability and abnormal gene expression. This has been shown in yeast and human studies where defective H3 K56 acetylation led to spontaneous DNA damage and sensitivity to genotoxic stress [8,9]. Moreover, while normal cells have dedicated control mechanisms for maintaining the steady-state levels of histone PTMs, a number of malignant cells have been shown to possess globally altered histone PTM patterns [10–12]. For example, abnormal levels of H3 K4 and H3 K79 methylations were found in a number of acute leukemias [13–15]. These studies suggest the critical roles histone PTMs play in the development and progression of cancer, but also indicates their potential as biomarkers and therapeutic targets for cancer diagnosis, monitoring and/or treatment.

The identification and quantification of histone PTMs is crucial to understand their molecular function in health and disease. Traditionally, analysis of histone PTMs is performed by immunochemical techniques, such as western blots or ELISAs, which requires the development of PTM- and peptide-specific antibodies. The disadvantages of these techniques are three-fold. First, the approach relies heavily on the prior knowledge of the type and position of the modification of interest. For example, antibodies can rarely be used to discover new sites of modification, but are rather used to detect and quantify a previously known PTM. Additionally, the specificity of antibodies against distinct sites and modifications remains a challenge due to significant cross-reactivity with other PTMs [16]. Finally, antibodies directed towards specific histone PTMs may fail to properly bind due to the presence of nearby PTMs disrupting or inhibiting their interaction [17]. These pitfalls reveal the need for unbiased approaches for quantifying histone PTMs.

Over the last decade, mass spectrometry (MS) has emerged as a powerful tool for characterization and quantitation of site-specific and global levels of histone PTMs. MS offers several advantages including reproducibility, specificity, and ability to rapidly measure numerous PTMs in a single experiment [18]. Previous work from our group and others has utilized both discovery and targeted approaches for the detection and quantitation of histones PTMs by bottom-up, middle-down, and top-down proteomics. These methods have been used to assess endogenous PTM turnover [19–21], to determine PTM dynamics throughout the cell cycle [22], and to examine changes in PTMs in disease states [23,24].

Our ability to reliably detect and quantitate histone PTMs by MS can be affected by various methodological choices. Sample preparation is one of the critical factors that can affect efficient detection of histone peptides via MS. Histone proteins exhibit unique chemical properties, which makes their isolation from biological samples a relatively simple task. Typically, for high throughput MS analysis histone proteins are isolated from a million or more cells, to yield sufficient material for multiple LC-MS/MS injections. However, this methodology is not applicable for the study of clinical samples with limited cells. Consequently, it is very important to assess the compatibility of existing histone assay methodologies and technologies for these types of applications. In the current study, we investigated the scalability of the input material by modifying the standard histone assay protocol to enable histone isolation from across a range of cell amounts. A translational application for use of this method is in clinical studies, where there may be limitations in the number of cells available for analysis. We demonstrated the scalability using histone proteins isolated from HeLa-S3 cells ranging in number

from $10^3$ to $10^6$ by eliminating nuclear isolation and reducing the number of propionylation steps (S1 Fig). Our results exhibit capability of the improved protocol to detect and quantify 61 histone peptides from 1,000 primary human bone marrow cells and 37 peptides in malignant cells from Acute Myeloid Leukemia (AML) patients. We anticipate that the new approach presented in this work will open a new frontier for MS-based applications in biomedical and translational research.

## Materials and methods

### Cell samples

HeLa-S3 cells were purchased from the National Cell Culture Center. Human bone marrow CD34$^+$ cells from healthy donors (NBMs) were purchased from Stemcell Technologies Inc. (Vancouver, Canada; n = 3). AML samples were obtained from the University of Pennsylvania (N = 3: AML1245, AML2093 and AML2373), and the ECOG-ACRIN group (N = 3 from NCT00046930: 5646646, 6815914 and 9600462). Patients provided informed consent according to the Declaration of Helsinki for collection and use of sample materials in IRB-approved research protocols at Weill Cornell Medicine, the University of Virginia and the University of Pennsylvania. Samples were subjected to Ficoll separation on the day of collection, and mononuclear cells were viably frozen. Cell were thawed at 37˚C in high-protein medium, treated with 1:10 volume of DNase (1mg/mL, Sigma-Aldrich, St. Louis, MO, USA), washed with PBS at 4˚C, and depleted of CD3$^+$ and CD19$^+$ cells using magnetic beads per manufacturer's recommendations (Miltenyi Biotec, Bergisch Gladbach, Germany). The myeloid enriched cells were washed in PBS at 4˚C and cell pellets were flash frozen on dry ice and stored at -80˚C until use.

### Histone extraction

Histone extraction was performed as described previously with some modifications [23]. Cell pellets were lysed directly with 0.2 M $H_2SO_4$ or with nuclear isolation buffer, NIB (15 mM Tris, 60 mM KCl, 15 mM NaCl, 5 mM $MgCl_2$, 1 mM $CaCl_2$, 250 mM sucrose, 0.4 mM 4-(2-aminoethyl)benzenesulfonyl fluoride hydrochloride, 10 mM sodium butyrate, 0.3% Nonidet P-40 and 1 mM dithiothreitol, pH 7.5). The nuclear pellet isolated with the latter approach was subsequently washed twice with NIB without detergent and resuspended in 0.2 M $H_2SO_4$, which extracts histones and other acid-soluble proteins. Trichloroacetic acid was added to a final concentration of 20% v/v to the supernatants that contained histones. The isolated histones were washed first with 0.1% HCl in acetone, washed twice with acetone, and then air-dried in a chemical fume hood.

### Histone propionylation and in-solution tryptic digestion

The propionylation reaction and tryptic digestion were adapted from previous works [20,25]. Histones were subjected to one or two rounds of propionylation before and after tryptic digestion. The dried histone pellet was resuspended in 5 μL of 0.1 M NH4HCO3 and mixed with 20 μL of freshly prepared 3:1 (v/v) isopropanol: propionic anhydride mixture. The solution was adjusted to pH 8 by adding multiple rounds of 5 μL aliquots of concentrated NH4OH solution (28% NH3 in water). The pH of the sample was measured at each addition of NH4OH by spotting 0.2 μL on a pH indicator strip. The sample was then incubated at 50 oC for 1 h for one round of propionylation or for 20 min for two rounds of propionylation. The sample was completely dried in a SpeedVac concentrator following each round of propionylation. After the derivatization, histones were digested with 0.5 μg of trypsin at 37˚C overnight.

Then, after an additional one or two rounds of propionylation, the digest was resuspended in 0.1% TFA prior to MS analysis.

## Proteomic analysis and data handling

Histone peptides and the standard peptide mix were analyzed on triple quadrupole MS (ThermoFisher Scientific TSQ Quantiva) directly coupled with UltiMate 3000 nano-LC system. Peptides were first loaded onto an in-house packed trapping column (3 cm × 150 μm) and then separated on New Objective (Woburn, MA) PicoChip analytical column (10 cm × 75 μm). Both columns were packed with Bischoff ProntoSIL C18-AQ, 3 μm, 200 Å resin (New Objective). The chromatography gradient was achieved by increasing percentage of buffer B from 0 to 35% at a flow rate of 0.300 μl/min over 45 minutes. Solvent A: 0.1% formic acid in water, and B: 0.1% formic acid in 95% acetonitrile. The instrument settings were as follows: collision gas pressure of 1.5 mTorr; Q1 peak width of 0.7 (FWHM); cycle time of 2 s; skimmer offset of 10 V; electrospray voltage of 2.5 kV. Targeted analysis of unmodified and various modified histone peptides was performed using collision-induced dissociation. All transitions (S7 Table) are previously published [19,20]

Raw MS files were imported and analyzed in Skyline software with Savitzky-Golay smoothing [26]. All Skyline peak assignments were manually confirmed. Peptides were considered to be detected if the signal to noise ratio was above 3 and quantifiable and there was at least one modification state detected in addition to the unmodified peptide. Peptide peak areas from Skyline were imported into an in-house built software to plot bar graphs representing relative abundances of each histone modifications. The relative abundances were determined based on the mean of biological replicates with error bars showing standard deviation. For each biological replicate, one technical replicate was acquired, with the exception of data acquired for $1 \times 10^6$ cells which had three technical replicates. Peptide modifications are abbreviated as un (unmodified), me1 (monomethyl), me2 (dimethyl), me3 (trimethyl), and ac (acetyl).

## Real-Time quantitative polymerase chain reaction

*SETDB1* mRNA levels was assessed by Real-Time Quantitative Polymerase Chain Reaction (RT-qPCR) using HPRT1 and ACTB as endogenous controls in AML specimens. Briefly, RNA was extracted using the AllPrep DNA/RNA Mini Kit (Qiagen, Hilden, Germany). Reverse Transcription was carried out at 42˚C for 1 hour using 100 ng of total RNA, M-MuLV Reverse Transcriptase (NEB, Ipswich, MA), and a 3:1 mixture of random hexamers (ThermoFisher), to Anchored Oligo-dT, (IDT, Coralville, IA). Subsequently, cDNA was diluted 10-fold and 3 μl of diluted cDNA was used in each 10 μl RT-qPCR reaction. RT-qPCR reactions were setup using 2x Hot-Start Taq Master Mix (NEB), EvaGreen Intercalating Dye (Biotium, Fremont, CA), and the following PrimeTime® Primers from IDT: SETDB1 (Hs. PT.58.25313824), HPRT1 (Hs.PT.58v.45621572), and ACTB (Hs.PT.38a.22214847). RT-qPCR was performed using the CFX384 Touch™ Real-Time PCR Detection System and CFX Maestro software (Bio-Rad, Hercules, California). Relative Quantitation of setdb1 mRNA was performed using the model defined in Pfaffl [27].

## Statistics

All values are reported as the mean ± one standard deviation. ANOVA was used with a Bonferroni correction to test for statistical significance at a p-value <0.05. Analysis of covariance (ANCOVA) was used to determine statistical significance in the coefficients of variation (CV).

## Results

### Effect of nuclear isolation step for low cell numbers

Typical histone extraction methods rely on nuclear extraction buffer containing mild, non-ionic detergents for cell lysis followed by successive washes and centrifugation for isolation of intact nuclei [25]. After every washing step, nuclei are pelleted by low-speed centrifugation (*i.e.*, 600 x *g*) and the supernatant is removed. Although this protocol has been successfully employed in a wide range of epiproteomic studies in our research group and others [21,23,28], its feasibility for quantifying the abundance of histone PTMs in low input samples has not been investigated. In the current study, we compared the MS-based quantification of histone samples obtained after acid extraction of isolated nuclei and after direct acid extraction of whole cells. Histone peptides generated from $1 \times 10^4$ and $5 \times 10^4$ HeLa-S3 cells were subjected to targeted LC-MS/MS. For samples with $1 \times 10^4$ cells, a significant increase in the extracted peak area was observed for 12 of the 89 quantifiable HeLa-S3 histone peptides in samples prepared without nuclear isolation (Fig 1A, S1 Table). No significant increases were observed at $5 \times 10^4$ cells (S2A Fig). The average peak area for all the peptides at $1 \times 10^4$ and $5 \times 10^4$ cells increased by 2.6- and 1.8-fold, respectively, with the omission of the nuclear isolation step. Furthermore, assessment of the coefficient of variation (CV) for each peptide at both $1 \times 10^4$ (Fig 1B) and $5 \times 10^4$ cells (S2B Fig) revealed that, while there is a significant difference in the variation of the CV values with and without the nuclear isolation, the samples without nuclear isolation show greater consistency across the dynamic range of peak areas (S3A and S3B Fig). These results suggest that elimination of the nuclear isolation step affords an increase in peptide peak area for low cell number samples without contributing to increased variance across samples.

### Reduced propionylation steps of histones improves MS signal

Propionylation of unmodified and monomethylated lysine residues is necessary to reduce digestion of lysine-rich histones by trypsin and creates consistent peptides for quantitation. Chemically, the addition of the propionyl group increase the hydrophobicity of the hydrophilic histone peptides, causing an increase in retention time to allow for adequate separation. The standard histone preparation protocol is time-consuming, typically requiring multiple days of sample preparation. Additionally, the protocol is prone to significant sample loss during pH measurement, undermining its compatibility for sample-limited applications. To limit sample loss, we compared the effects of one versus two rounds of histone propionylation from $1 \times 10^6$ HeLa-S3 cells. The result showed a significantly higher signal intensity for 56 of the 89 histone peptides which were quantifiable with one round of propionylation compared to two rounds (Fig 2A, S1 and S2 Tables). On average, there was a 4.2-fold increase in the extracted peak area of peptides detected with one round of propionylation. Further, quantitation of histone PTMs for histone H3.1 K27 and K36 show no differences in the relative abundance of each modification state between one and two rounds of propionylation (Fig 2B). Some peptides, such as H3.3 K27ac K36un, were observed in samples with one round of propionylation but not with two rounds (S4 Fig). These results show that one round of propionylation contributes to an overall signal increase while still yielding similar results for most PTMs.

### Assessment of the scalability of input material for histone assay

By combining the methodological improvements noted above, we next sought to assess the effects of relative measurements on fewer cells. Serial dilutions of HeLa-S3 cells down to $1 \times 10^3$ cells were directly acid extracted to isolate histones and subjected to one round of propionylation before and after tryptic digestion. Resulting peptides were resuspended in equal volumes

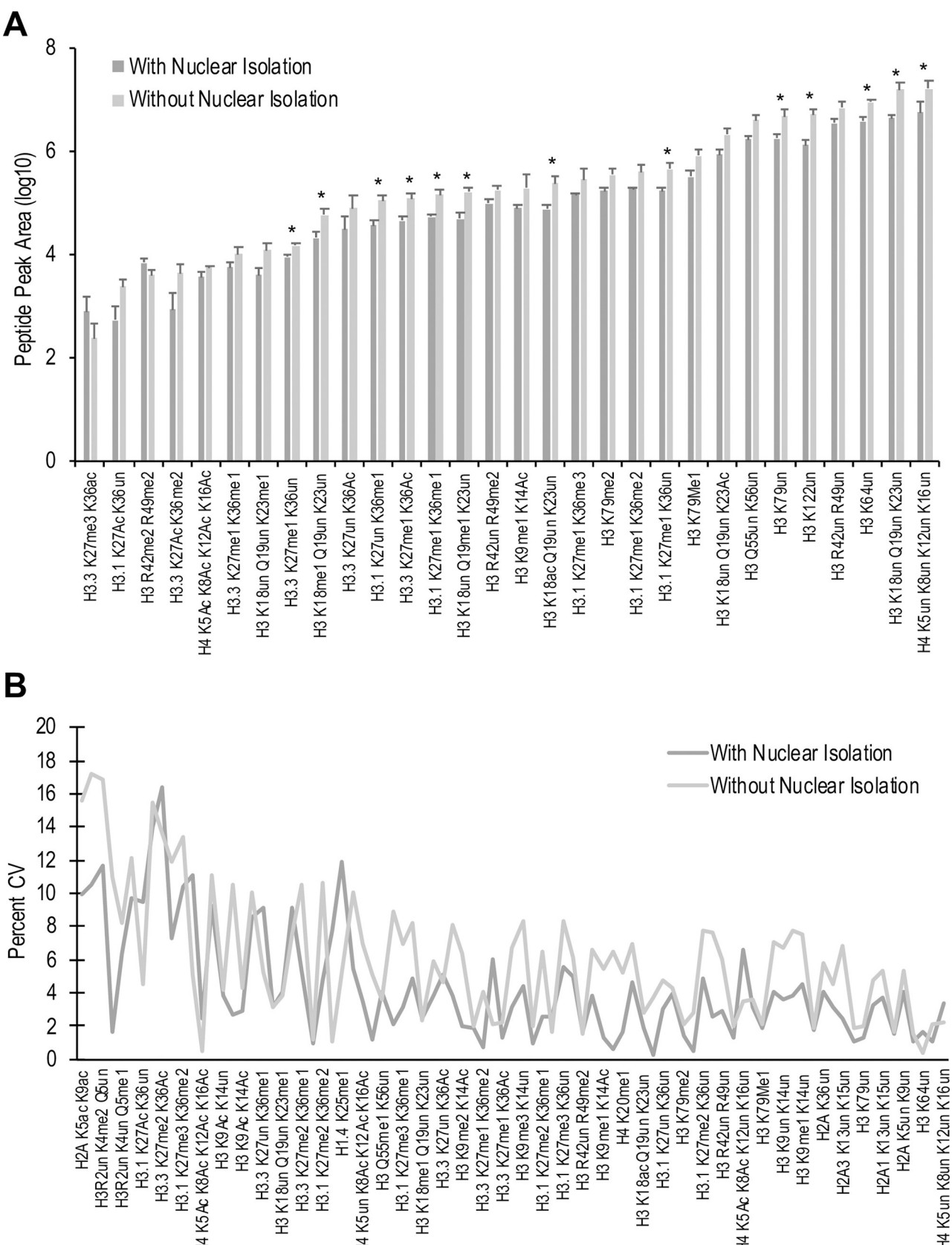

**Fig 1. Histone peptide detection with or without nuclear isolation.** Comparison of (A) extracted peak area of histone peptides and (B) percentage CV values of 29 representative histone peptides from $1 \times 10^4$ cells with or without nuclear isolation step. Error bars represent the standard deviation of biological triplicates. Asterisks (*) represent statistical significance at a p-value <0.05 using ANOVA with a Bonferroni correction.

and equal injection volumes were loaded onto the analytical column for separation followed by targeted LC-MS/MS analysis. Comparison of the peak intensity shows an overall ~1.4 fold-change in signal between $1 \times 10^3$ and $1 \times 10^6$ cells for both H3.1 K27un K36me2 (Fig 3A and 3B)

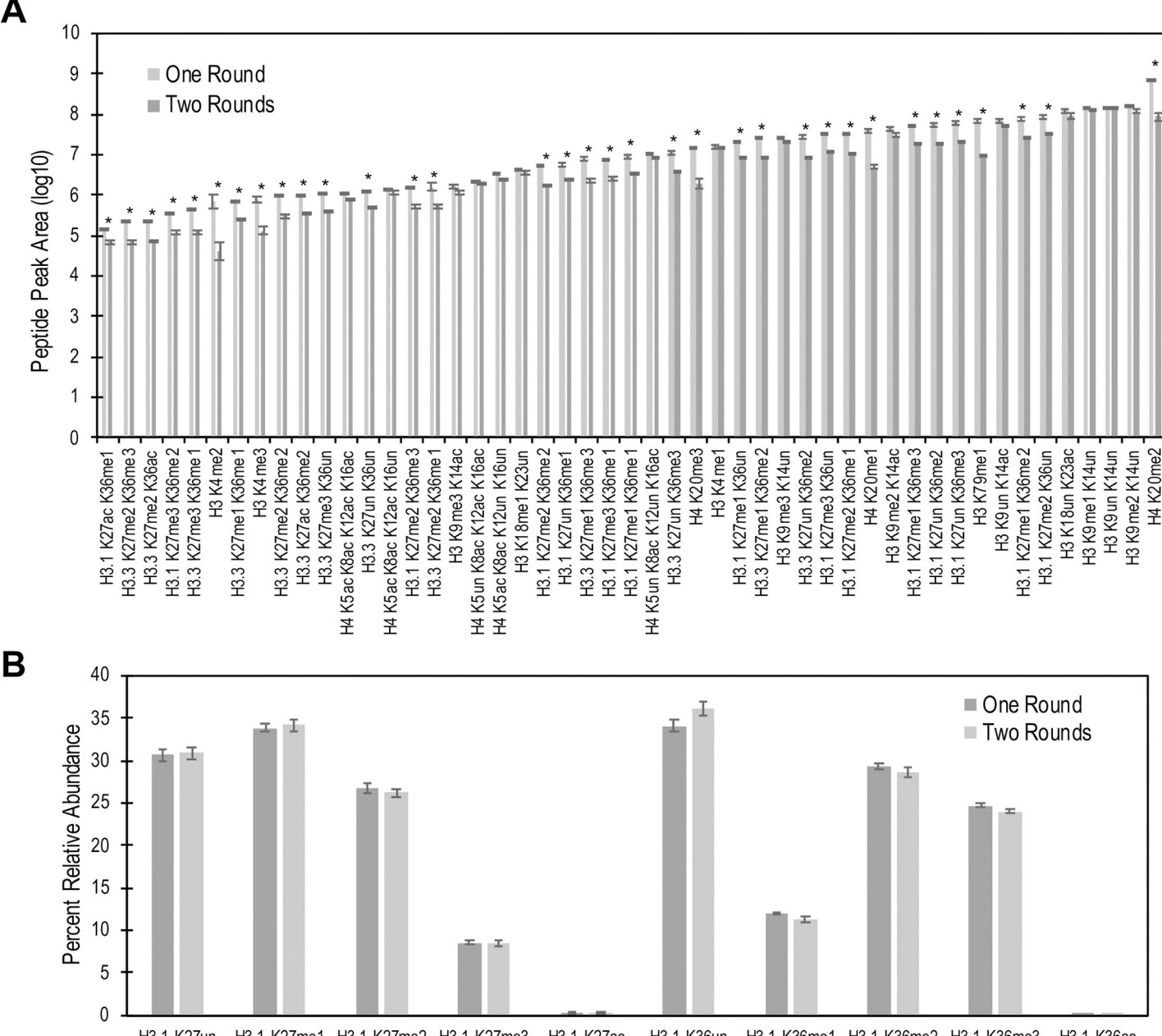

**Fig 2. Peptide peak area comparison of one versus two rounds of histone propionylation.** (A) Extracted peak area of one vs two rounds of propionylation for representative histone marks. Histones were acid-extracted from $10^6$ HeLa cells and subjected to one or two rounds of propionylation (N = 4). (B) Bar graphs showing the relative abundances of histone marks. Both propionylation strategies result in comparable relative abundances. Asterisks (*) represent statistical significance at a p-value <0.05 using ANOVA with a Bonferroni correction.

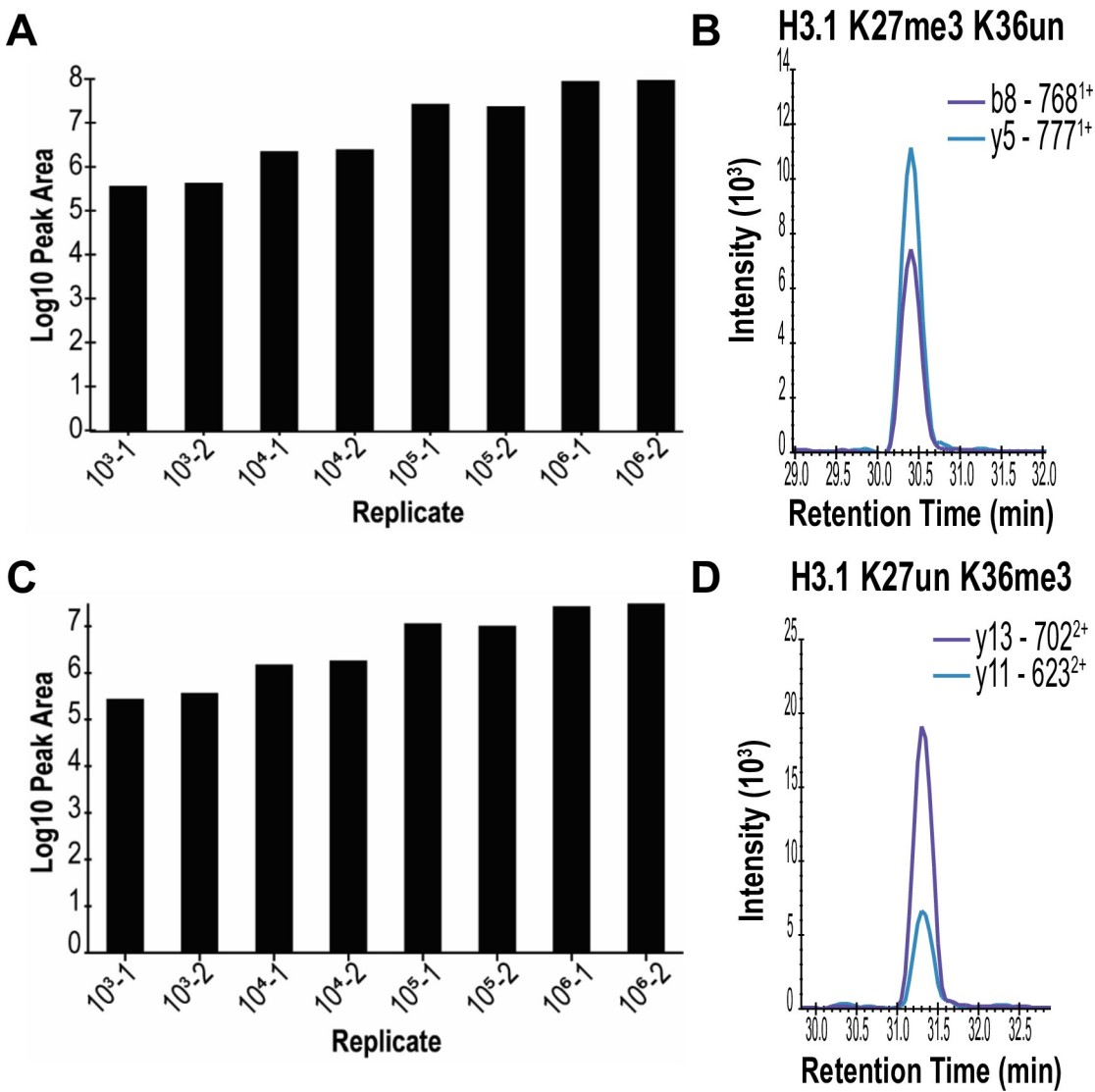

**Fig 3. Lowering the input cell amount for assessment of histone marks.** HeLa-S3 cells were diluted to obtain a range between $10^6$ and $10^3$ cells. Following preparation, all samples were resuspended in equal volumes and injected in equal amounts for targeted LC-MS/MS. Peak area comparison of representative histone marks generated from $10^3$–$10^6$ HeLa-S3 cells for (A) H3.1 K27un K36me3 and (C) H3.1 K27me3 K36un. Representative chromatograms are shown on the right of each panel for (B) H3.1 K27un K36me3 and (D) H3.1 K27me3 K36un. Samples are shown as the number of cells and the replicate number.

and H3.1 K27me3 K36un (Fig 3C and 3D). Of the 89 histone peptides quantifiable at $1\times10^6$ HeLa-S3 cells, 61 peptides were quantifiable at $1\times10^3$ cells (Table 1 and S3 Table).

## Determination of histone modifications from primary human cells

To assess if the modified method is translatable to primary human specimens, we first applied the approach to human CD34$^+$ progenitor cells obtained from bone marrow of healthy donors (NBMs). As these represent rare cells in the bone marrow, a method geared towards low cell number inputs would benefit the study of histone PTMs present in this cell type and other rare populations. Using the modified method with an input of $1\times10^3$ cells, all peptides detected in HeLa cells were also detected in NBMs, along with one additional peptide, H3 K9ac K14ac

**Table 1. Peptides quantified at 1000 cells.**

| Peptide | HeLa | NBMs | AML | Peptide | HeLa | NBMs | AML |
|---------|------|------|-----|---------|------|------|-----|
| H4 K5un K8un K12un K16un | X | X | X | H3 K9un K14un | X | X | X |
| H4 K5un K8un K12un K16ac | X | X | X | H3 K9un K14ac | X | X | X |
| H4 K5un K8un K12ac K16ac | X | X |   | H3 K9me3 K14un | X | X |   |
| H4 K5un K8ac K12un K16ac | X | X | X | H3 K9me3 K14ac | X | X |   |
| H4 K5un K8ac K12ac K16ac | X | X |   | H3 K9me2 K14un | X | X | X |
| H4 K5ac K8un K12un K16un | X | X | X | H3 K9me2 K14ac | X | X | X |
| H4 K5ac K8ac K12un K16un | X | X |   | H3 K9me1 K14un | X | X | X |
| H4 K5ac K8ac K12ac K16un | X | X |   | H3 K9me1 K14ac | X | X | X |
| H4 K5ac K8ac K12ac K16ac | X | X |   | H3 K9ac K14ac |   | X |   |
| H3.1/2 K27un K36un | X | X | X | H3.3 K27un K36un | X | X |   |
| H3.1/2 K27un K36me3 | X | X | X | H3.3 K27un K36me3 | X | X | X |
| H3.1/2 K27un K36me2 | X | X | X | H3.3 K27un K36me2 | X | X | X |
| H3.1/2 K27un K36me1 | X | X | X | H3.3 K27un K36me1 | X | X |   |
| H3.1/2 K27me3 K36un | X | X | X | H3.3 K27un K36ac | X | X |   |
| H3.1/2 K27me3 K36me2 | X | X |   | H3.3 K27me3 K36un | X | X | X |
| H3.1/2 K27me3 K36me1 | X | X | X | H3.3 K27me3 K36me1 | X | X |   |
| H3.1/2 K27me2 K36un | X | X | X | H3.3 K27me2 K36un | X | X | X |
| H3.1/2 K27me2 K36me3 | X | X |   | H3.3 K27me2 K36me2 | X | X |   |
| H3.1/2 K27me2 K36me2 | X | X |   | H3.3 K27me2 K36me1 | X | X |   |
| H3.1/2 K27me2 K36me1 | X | X | X | H3.3 K27me1 K36un | X | X |   |
| H3.1/2 K27me1 K36un | X | X | X | H3.3 K27me1 K36me3 | X | X | X |
| H3.1/2 K27me1 K36me3 | X | X | X | H3.3 K27me1 K36me2 | X | X | X |
| H3.1/2 K27me1 K36me2 | X | X | X | H3.3 K27me1 K36me1 | X | X |   |
| H3.1/2 K27me1 K36me1 | X | X | X | H3.3 K27me1 K36ac | X | X |   |
| H3 K79un | X | X | X | H3 K18un K23un | X | X | X |
| H3 K79me1 | X | X | X | H3 K18un K23ac | X | X | X |
| H3 K79me2 | X | X | X | H3 K18ac K23un | X | X | X |
| H4 K20un | X | X |   | H3 K18ac K23ac | X | X | X |
| H4 K20me1 | X | X | X | H3 K4un | X | X |   |
| H4 K20me2 | X | X | X | H3 K4me1 | X | X |   |
| H4 K20me3 | X | X |   |  |  |  |  |

HeLa, NBMs, and AML specimens ($10^3$ cells each) were prepared according to the modified method and analyzed by LC-MS/MS. Quantifiable peptides for each cell types are marked with an X.

(Table 1, S4 Table and S5 Fig). Representative chromatograms show defined peaks for H3 K9me1 K14ac, H4 K5un K8un K12un K16ac, H3 K18un Q19un K23ac, H3 K79un, H3.1/2 K27me1 K36me2, and H3.3 K27me1 and K36un (Fig 4).

Finally, we determined the abundance of histone PTMs in malignant cells from AML patients. We used the revised protocol to determine the relative abundance of histone PTMs in six AML patients at $1 \times 10^6$ cells. Levels of H3 K9me2 and H3 K9me3 were high in a subset of three AML patients while low in the second subset of three AML patients (Fig 5A and S5 Table). We considered the possibility that the difference in H3K9 methylation levels could be associated with differences in expression of SETDB1, a histone lysine methyltransferase of the H3 K9 residue [29]. We performed quantitative PCR to assess for mRNA transcript levels of SETDB1 in the AML specimens. We observed a difference in the relative mRNA transcript levels of SETDB1 between the groups (S6 Fig). SETDB1 mRNA was higher on average in patients

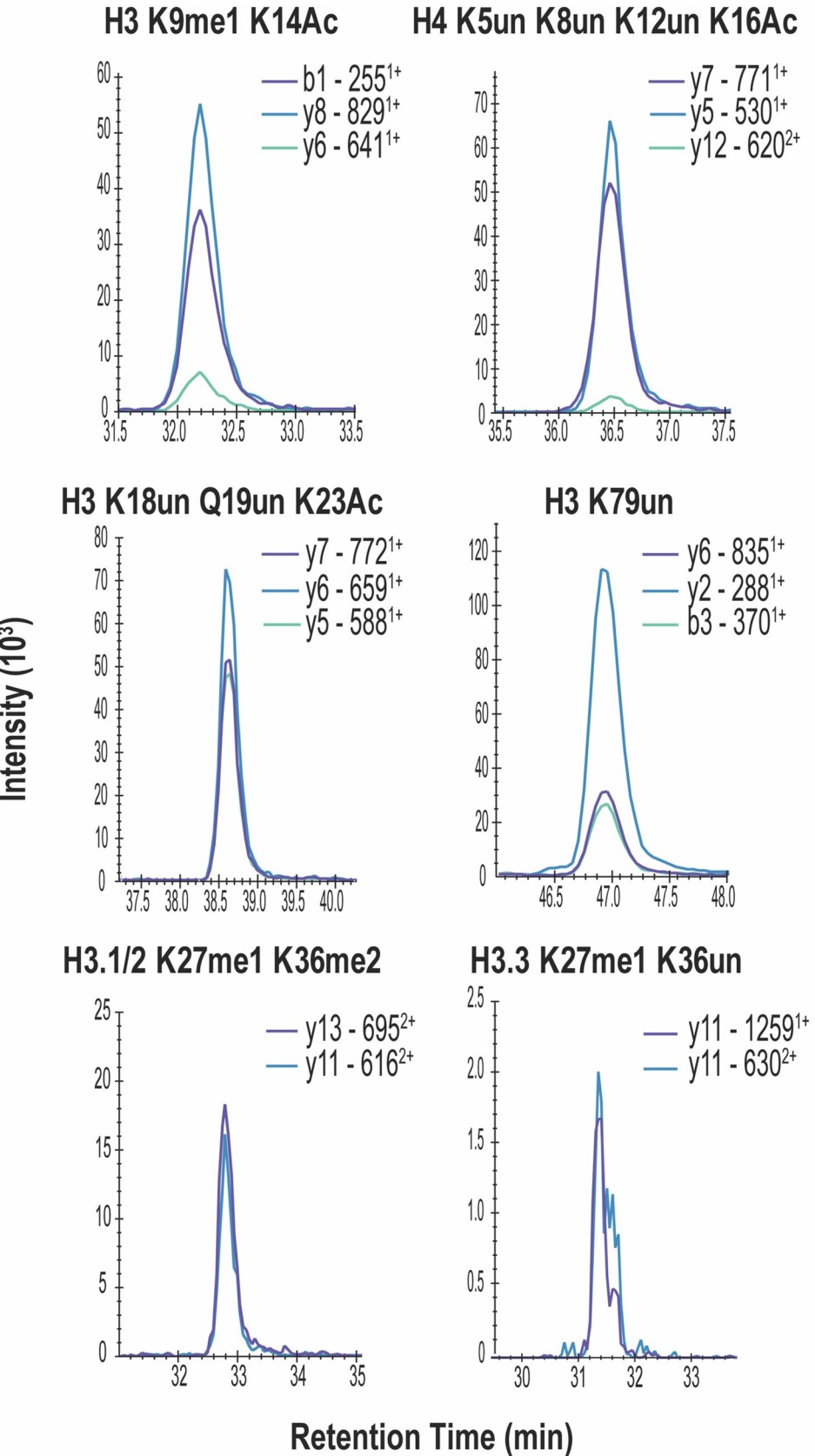

**Fig 4. Quantitation of histone modifications from 1000 primary human NBMs.** Representative chromatograms of modified histone peptides from $10^3$ human normal bone marrow CD34$^+$ cells (N = 3).

with increased H3 K9me2/3, suggesting a regulatory role for SETDB1 and/or H3 K9 methylation levels in the AML specimens investigated. Next, we assessed for the detection of histone PTMs in $1\times10^3$. Cells from three AML patients were prepared according to the modified method and 37 peptides were quantifiable in the proteomic analysis from $1\times10^3$ malignant cells (Table 1 and S6 Table). Representative chromatograms of modified histone peptides are plotted in Fig 5B.

## Discussion

In this study, we assessed the feasibility of reducing the sample processing steps for preparation of histone peptides for targeted LC-MS/MS to enable analysis of very low cell number samples. Prior work by Garcia et al. [25] is considered the gold standard for sample preparation of histones for bottom-up analysis. While this protocol provides reliable results from little at 1 μg of protein, those amounts may be difficult or impossible to obtain for some clinical and translational applications. Further advancement of the method into the "one-pot" approach eliminated the need for off-line purification of histones [30], opening the door to the possibility of reduced sample amounts due to reduced sample loss during individual histone isolation.

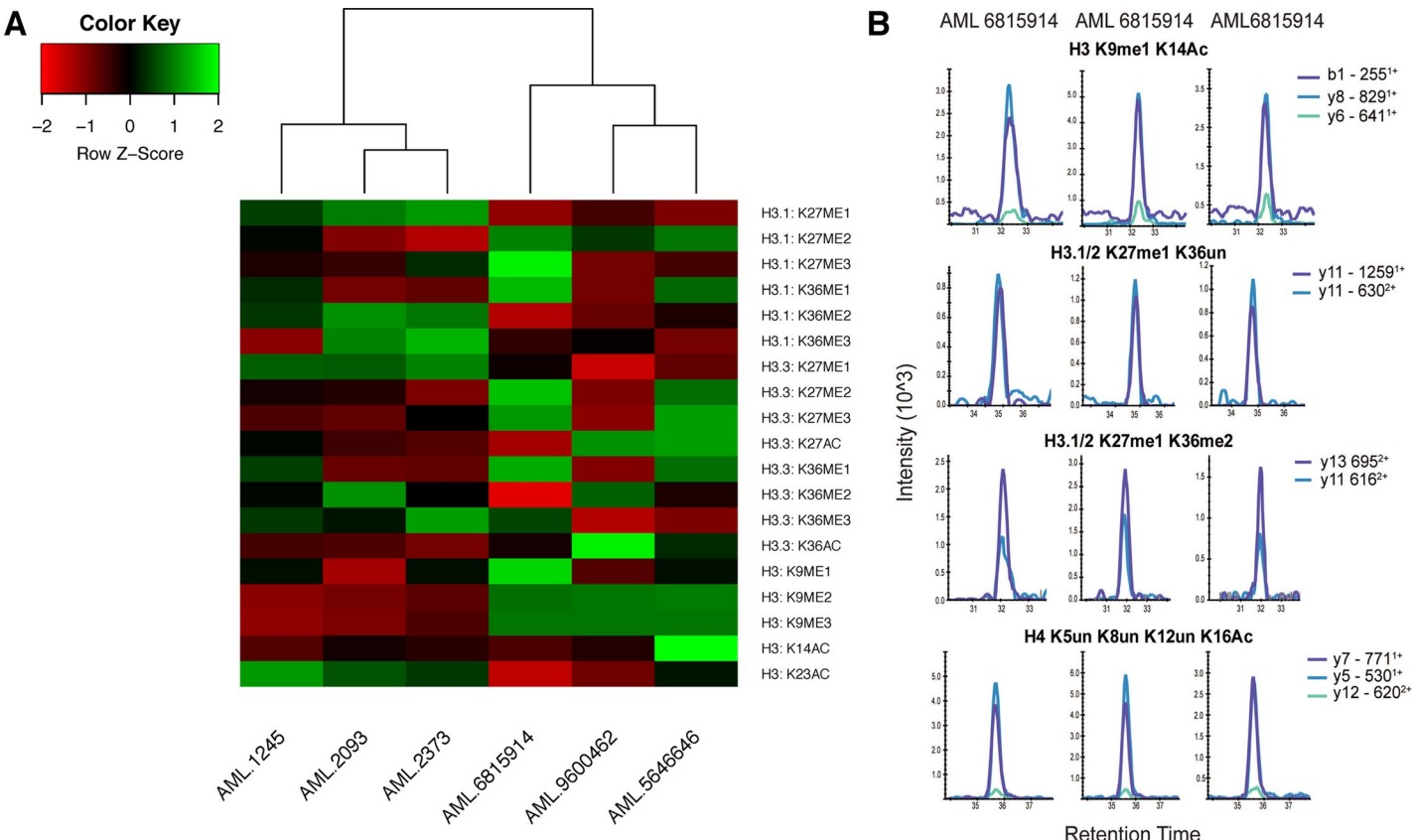

**Fig 5. Histone PTMs in AML patient specimens.** (A) Heatmap of relative histone modification levels detected in $10^6$ cells. (B) Histones were extracted directly by acid extraction from $10^3$ AML patient cells and subjected to one round of propionylation before and after tryptic digestion. Shown are representative chromatograms of modified histone peptides.

There is great interest in understanding histone PTM profiles in rare cell populations, including malignant and non-malignant sources, for both basic and clinical research. A critical need for the analysis of clinical samples is the ability to obtain high quality data from low sample inputs. Indeed, analysis of histones from low cell numbers has been the focus of multiple publications in recent years. In 2018, Gou et al. [31] was able to analyze histones from 50,000 cells from both cell lines and primary cells using low resolution mass spectrometry. This represents a significant reduction in the number of cells needed for quantitative analysis of histone modifications compared to prior publications.

Additional methodological baselining has been performed on clinical specimens, both from primary cells and patient tissues. Recent work from Noberini et al. [32] established $5 \times 10^5$ cells as a reliable amount for quantitation of histone modifications from primary cells, translating to less than 10 µg of histone octomers. Similar cell numbers are needed for histone analysis from FFPE tissue collected by laser capture microdissection through the pathology tissue analysis of histones by mass spectrometry approach, or PAT-H-MS [33]. These low histone amounts have also been applied to normal and tumor specimens, establishing that 2–5 µg of histone proteins are needed to obtain quantifiable results.

The approach we outline here is a further improvement to the above referenced methods to enable quantitation of histone PTMs from as low as 1000 primary cells. Exclusion of the nuclear isolation step resulted in a 2.6-fold improvement in the detection of several histone PTMs, with significant advantages seen at very low cell numbers (Fig 1A). Reducing the number of propionylations resulted in additional improvements in signal intensity (Fig 2A). It is important to note, however, that the revised protocol performs better for low cell number samples and fewer advantages are seen at higher cell numbers. At 50,000 cells, the amount used by Guo et al. [31], no benefit was observed when using our modified protocol (S1A Fig). Use of the traditional preparation methods, with nuclear isolation and increased propionylations, may be more applicable for larger starting amounts.

This research aids to expand the scope of "multi-omics," especially with the focus on cancer. Single cell genomic applications are regularly used, such as single cell RNA-Seq [34] and single nucleus RNA-Seq [35]. The field of metabolomics is also geared towards very low cell numbers. Metabolomics is capable of analyzing 100 breast cancer cells [36] to as low as single cells for specific metabolites, such as glucose phosphate [37]. Lipidomics has been applied to fewer than 100 cells with LC-ESI-MS/MS [38]. High quality data from low cell numbers have been difficult to acquire by LC-based proteomics, but advancements in instrument sensitivity along with modifications in sample preparation as outlined here can provide useful results to better align with the capabilities of other "multi-omic" approaches.

One area where this approach can be readily utilized is the profiling of hematologic malignancies. Multiple aberrations in histone PTMs have been observed in leukemias and lymphomas, such as EZH2 mutations altering H3 K27 methylation [39], DOT1L dysregulation of H3 K79 methylation with MLL translocations [40], and KMT2D mutations affecting H3 K4 methylation [41]. Detection of histone PTMs could not only be applicable to disease prognosis as shown in solid tumors [11,42], but could also provide insight about potential biological mechanisms of disease facilitated by epigenetic mediators. Within AML, epigenetic modifiers are known to play a role in the pathogenesis of disease in a subset of AML patients, with more than a dozen enzymes dysregulated in AML contributing to alterations in histone PTMs [43].

Application of our revised method to AML patient specimens showed a difference in H3 K9me3 across two subsets of AML samples (Fig 5). This difference in H3 K9me3 was consistent with the mRNA levels of SETDB1 (S6 Fig), the enzyme responsible for trimethylation of H3 K9, suggesting that the loss of SETDB1 results in the reduction of H3 K9me3. Alterations in H3 K9me3 have previously been seen in AML, showing a reduction in this modification

specifically in promoter regions [44]. Recently, SETDB1 was shown to play a significant role in survival in AML, with patients exhibiting higher SETDB1 mRNA levels experiencing increased survival times (median of 26.3 months in SETDB1 high vs 9.5 months in SETDB1 low) [45]. The limited data presented suggests an approach to identifying patients from limited materials with aberrant H3 K9me3 and associated SETDB1 expression which may contribute to disease pathogenesis and survival. It is important to note, however, that the data in this field are still developing and there is no clear consensus at to the role of H3 K9me3 or SETDB1 in AML. While the underlying biology is still under investigation, our data show that a revised sample histone preparation can be applied to sparse clinical samples to better understand how histone PTMs are altered across individuals.

## Conclusions

Sample limitations represent a significant challenge for the translation of proteomics to the clinic and methods need to be modified to utilize this material. We have described a modified method for quantifying histone modifications from samples with limited material. This approach may allow use of residual samples from clinical diagnostics for research, and/or rare cell populations, to assess for potential disease biomarkers and hypotheses related to mechanisms of disease.

## Supporting information

**S1 Fig. Schematic overview comparing the standard preparation protocols with the modified procedure.**
(TIF)

**S2 Fig. Elimination of nuclear isolation in $5 \times 10^4$ cells.** Comparison of (A) extracted peak area of histone peptides and (B) percentage CV values of histone peptides from $5 \times 10^4$ cells with and without the nuclear isolation step. Error bars represent the standard deviation of two instrument replicates within a single experiment.
(TIF)

**S3 Fig. Variation in CV with and without nuclear isolation.** Log10 of peak area for each cell number was plotted with log10 of the coefficient of variation (CV). (A) $1 \times 10^4$ cells show the same slope with (red) and without (green) nuclear isolation and a statistically significant difference in the lines (p = 6.055e-6), but not the slopes (p = 0.4254) by analysis of covariance (ANCOVA). (B) $5 \times 10^4$ cells show different slopes with (red) and without (green) nuclear isolation and a statistically significant difference in the lines (p<2.2e-16) and slopes (p = 0.001876) by ANCOVA.
(TIF)

**S4 Fig. Reduction in propionylations rounds reveals peptides from $1 \times 10^4$ cells.** Skyline extracted peaks for H3.3 K27ac K36un shows the presence of the peptide in one round of propionylation and absent in two rounds.
(TIF)

**S5 Fig. Modified protocol reveals H3 K9K14 peptide at $1 \times 10^3$ cells.** Comparison of H3 K9ac K14ac peptide in HeLa (left) and NBM (right) shows the presence of the peptide in NBM but not HeLa.
(TIF)

**S6 Fig. Correlation of histone modifications with RNA transcript levels.** Relative *SETDB1* mRNA levels as determined by qPCR in three samples with low H3K9me2/3 (red) and three

samples with high H3K9me2/3 (blue).
(TIF)

**S1 Table. Percent relative abundance of histone PTMs with and without nuclear isolation prior to histone extraction.**
(XLSX)

**S2 Table. Percent relative abundance of histone PTMs with one or two rounds of propionylation.**
(XLSX)

**S3 Table. Percent relative abundance of histone PTMs at a range of HeLa cell numbers.**
(XLSX)

**S4 Table. Percent relative abundance of histone PTMs in CD34+ normal bone marrow.**
(XLSX)

**S5 Table. Percent relative abundance of histone PTMs in primary AML patient specimens at 1,000 cells.**
(XLSX)

**S6 Table. Percent relative abundance of histone PTMs in primary AML patient specimens.**
(XLSX)

**S7 Table. Complete list of transitions used in data acquisition.**
(XLSX)

## Acknowledgments

The authors thank Richard LeDuc for helpful discussions on this work.

## Author Contributions

**Conceptualization:** Nebiyu A. Abshiru, Jacek W. Sikora, Philip D. Compton, Paul M. Thomas, Neil L. Kelleher.

**Formal analysis:** Nebiyu A. Abshiru, Jacek W. Sikora, Jeannie M. Camarillo, Francine E. Garrett-Bakelman.

**Funding acquisition:** Francine E. Garrett-Bakelman, Neil L. Kelleher.

**Investigation:** Juliette A. Morris, Tak Lee.

**Methodology:** Nebiyu A. Abshiru, Jacek W. Sikora, Paul M. Thomas.

**Resources:** Yaseswini Neelamraju, Samuel Haddox, Caroline Sheridan, Martin Carroll, Larry D. Cripe, Martin S. Tallman, Elisabeth M. Paietta, Ari M. Melnick.

**Supervision:** Philip D. Compton, Paul M. Thomas, Francine E. Garrett-Bakelman, Neil L. Kelleher.

**Validation:** Jeannie M. Camarillo.

**Writing – original draft:** Nebiyu A. Abshiru, Jeannie M. Camarillo.

**Writing – review & editing:** Jeannie M. Camarillo, Paul M. Thomas, Francine E. Garrett-Bakelman, Neil L. Kelleher.

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
