## [Decision Letter · Decision Letter 0]

11 Jun 2020

PONE-D-20-14092

Targeted detection and quantitation of histone modifications from 1,000 cells

PLOS ONE

Dear Dr. Kelleher,

Thank you for submitting your manuscript to PLOS ONE. After careful consideration, we feel that it has merit but does not fully meet PLOS ONE’s publication criteria as it currently stands. Therefore, we invite you to submit a revised version of the manuscript that addresses the points raised during the review process.

The reviewers and felt that the study is worthwhile and interesting. However, a number of issues were raised that need to be addressed in the revised manuscript. I agree with their assessments.

We look forward to receiving your revised manuscript.

Kind regards,

C. Michael Greenlief, Ph.D.

Academic Editor

PLOS ONE

Journal Requirements:

2. Please amend both your Ethics Statement and Methods section to state what type of patient consent was granted (i.e., written, verbal, etc.)

Additional Editor Comments (if provided):

There are issues raised by reviewers 1 and 3, particularly with regards to Figure 5 and discussion of its contents.

Reviewers' comments:

Reviewer's Responses to Questions

**Comments to the Author**

1. Is the manuscript technically sound, and do the data support the conclusions?

Reviewer #1: Partly

Reviewer #2: Yes

Reviewer #3: Partly

2. Has the statistical analysis been performed appropriately and rigorously? 

Reviewer #1: Yes

Reviewer #2: Yes

Reviewer #3: I Don't Know

3. Have the authors made all data underlying the findings in their manuscript fully available?

Reviewer #1: Yes

Reviewer #2: Yes

Reviewer #3: Yes

4. Is the manuscript presented in an intelligible fashion and written in standard English?

Reviewer #1: Yes

Reviewer #2: Yes

Reviewer #3: Yes

5. Review Comments to the Author

Reviewer #1: The paper entitled "Targeted detection and quantitation of histone modifications from 1,000 cells" from Kelleher’s group is an interesting and well-planned research on the improvement of sample preparation for histone PTM study. The manuscript is good as for contents, methodology and data analysis, the writing style and the clarity of the exposition are accurate. Conclusions are clear and Ref list is complete and correct. I have no specific concern but it is mandatory to explain/correct the manuscript in some points:

· Fragment type (CID or HCD) should be provided in the method part.

· Authors claimed that serial dilutions were used to make sure HeLa-S3 cells down to 1×103 cells. Why don’t use flow cytometry here?

· Heatmap in the Figure 5A is missing. b and y ions should be labeled in the Figure 5B.

· b and y ions in the Figure 4- are these from unique peptides? Any evidences to support the localization of Me/Ac on the K? Representative MS2 should be provided here.

Reviewer #2: The author in this paper primarily demonstrated the detection of histone peptides from limited number of cells using a targeted LC-MS/MS approach. This paper is interesting and certainly has potential applications towards proteomics at few or even single cell level. Combined with previous publications such as Luo et. Al., Anal. Chem. 2017, 89, 21, 11664–11671, it becomes feasible to pursue multi-omics research with under 1000 cells, which will potentially resolve unique properties of rare population of cells. In this regard, I think this paper is significant, and meets the journal standard for publication, although it lacks originality (i.e., modified protocols are reported).

Additionally, it would be helpful for the author to have a discussion on how this work relates to current multi-omics research, and further demonstrate its capability to study particular type of cells of interest.

Reviewer #3: In this manuscript Abshiru et al. present work that improves the sensitivity on a number of cells analyze basis for proteomic quantitation of histone PTMs. The key step is skipping nuclei isolation for cell numbers less than 1x10^4 and disposing of additional rounds of derivitization. This makes significant advances in sensitive histone proteomic analysis for limited samples in a variety of applications. The overall quality of the work is very good if spotty in places and I am confident in its publication in the near future. There are a few issues that unfortunately preclude its expedited publication without an additional round of review.

Major issues:

1) Figure 5 is missing and has not been peer reviewed. This is the crucial data conveying any actual biological insight. These results are also poorly described in text and no statistical analysis is presented for the interpretation and focus on specific results.

2) The manuscript fails to put this work in proper context from an analytical perspective. The improvement is compared relative to a 13 year old paper with no mention of improvements or comparison to recent work.

3) The discussion of the biological results is near-completely absent and is lacking context as well.

4) There are some major issue around the statistical interpretation of figure 1, given the apparent use of instrument technical replicates on single samples to infer significance of differences between the workflows.

Statistics:

The authors are generally clear and transparent about the use of statistics; however, there is some questionable apples-to-oranges interpretation of the statistics.

In text comments:

88

“However, this methodology is not applicable for the study of clinical samples with limited cells.” – This claim is overstated. A million cells or even a few million cells are a feasible number for many clinical applications. Yes, this is true for some, or even many, clinical applications as well as some basic science questions. Certainly, sensitivity gains are broadly useful.

202

The data does not support these statements. Unless I missed something, the variances used for the statistics are technical and thus do not infer a statistically meaningful difference between the workflows. These results say that your instrument variance is smaller than the difference between these two samples. Repeating the workflow to measure the workflow variance would be necessary to make these statements. I think you could make more general claims of similarity. They certainly look similar and the data has not been obviously compromised by the new workflow.

219

“The standard histone propionylation protocol is time-consuming, typically requiring multiple days of sample preparation.” – Days, really? This only takes minutes of actual work and perhaps a few hours overall. Perhaps you are referring to from cells to the mass spec?

258

The use of the word “asses” is inappropriate.

274

“Histone PTMs’ assessment in 1×106 cells was performed in six patients.” – Rewrite this sentence

275

“We identified consistent levels of H3 K9me2 and H3 K9me3 between two patient groups (Fig 5A and S5 Table).” – Figure 5 does not appear to be the correct figure to cite here. Also this sentence is followed by “We considered the possibility that the difference in H3 K9me levels could be…” The meaning of these two sentences is unclear. “consistent between two patient groups” and “the different in…” do not follow each other. Certainly there is some point that is not clear. This is a significant crux of the paper regarding any biological insight and it is completely unclear.

286

Again, there is a misplaced figure. I see no heat maps in any of the main figures. This makes it hard to review the results.

291

Table 1 is titled as “identified” but the caption calls them “quantifiable”. There is a difference.

299

The paper cited is certainly important. It is an early effort that made important methodological innovations for histones. However, the use of the Gold standard moniker here seems misguided. Perhaps it is in the sense that it set a marker down of 1ug and is a good point of comparison in the distant past. Making a comparison to this work with respect to sensitivity is misleading. Dramatic improvements in sensitivity have been made in the intervening 13 years. Notably the introduction of the one-pot approach just a couple years later, which skips the HPLC purification step and analyzes the entire acid soluble fraction collectively, as is common today, dramatically improved the sensitivity. (J Proteome Res. 2009 Nov;8(11):5367-74) Instruments have also improved in the intervening years. It is true that good metrics of sensitive and efficient analysis are frequently notably absent there are a few more recent papers that are good points of comparison. There is already work down to 50,000 cells (Journal of Proteome Research, 20 Nov 2017, 17(1):234-242) and they did perform nuclei isolation. However, they do not state the amount loaded on column directly; however, there is recent top down work on histones the clearly states injecting 55 ng on column (J. Am. Soc. Mass Spectrom. 2019, 30, 12, 2548–2560). Clearly, the number to compare to in a modern context is much lower than 1µg. Also clear from looking for good points of comparison is that more work is needed that endeavors to bring these numbers further down with clearly defined metrics. Your work clearly does this but could use well researched context. It is your position as author to do this sort of research and educate the reader. As you fairly point out there is also a difference between on column sensitivity and the sample requirements.

325

This is an incredibly sparse discussion of the AML results. There is plenty of context to give here. It does not need to be really big but this is almost nothing.

Figure 1

The use of instrument technical variance does not support there being any statistical significance to the difference between with and without nuclei isolation, which is the point of this figure to my understanding.

Figure 3

The figure would be clearer with additional labels on A and C.The figure caption is slightly confusing primarily due to the organization and labeling. although I think I figured it out. I might go with just an A panel (for A & B) and a B panel (for C & D) if you are not describing the panels separately.

Figure 4

The caption oddly starts out on the very general subject before describing what the figure is.

Figure 5

Wrong figure included in file. This is a critical figure and I do not have access to critically review it.

Supplemental figure 1

It seems to me that this should be Figure 1 and not a supplemental figure. At the least it would be useful and informative to compare your results to previous work in the area of Leukemia and Histones PTM quantitation, particularly in vivo: (e.g. Blood (2014) 124 (21): 2202 & Blood (2019) 134(24): 2183-2194) or even in vitro J. Am. Soc. Mass Spectrom. 2004;15(1):77–86. Has this (*whatever this is* since the figure is missing and the results poorly described) been observed previously or is it novel. Does SETDB1 have an established role in leukemogenesis? A quick google search is a resounding yes, yet no mention here. I infer from Fig S6 that you probably observed higher H3K9me3. Contrary to the SETDB1 story I also see in the literature this biology is mostly connected with higher levels of acetylation, including K9ac which block SETDB1 activity and is antagonistic of K9me3. How to rationalize these.

There are a number of citations that worthy of inclusion (incomplete list):

From the Garcia Lab:

One-Pot Shotgun Quantitative Mass Spectrometry Characterization of Histones. J Proteome Res. 2009 Nov;8(11):5367-74

Dramatically improved sensitivity and established the practice of skipping further purification after dervitization, which is an integral part of your approach. Your work is an extension of gaining sensitivity by reducing steps.

From the Turner Lab:

Reading Signals on the Nucleosome With a New Nomenclature for Modified Histones. Nat Struct Mol Biol. 2005 Feb;12(2):110-2. doi: 10.1038/nsmb0205-110.

This established the Brno nomenclature for histone modifications used throughout this work. It would be helpful to reference this at line 164.

The Tiziana Bonaldi lab has published extensively on clinical applications of histone PTM proteomics on limited samples, including human tumors etc.

6. PLOS authors have the option to publish the peer review history of their article (what does this mean?). If published, this will include your full peer review and any attached files.

Reviewer #1: No

Reviewer #2: No

Reviewer #3: No

---

## [Author Response · Author response to Decision Letter 0]

24 Jul 2020

The Response to Reviewers can be found in the uploaded attachments.

---

## [Decision Letter · Decision Letter 1]

11 Aug 2020

PONE-D-20-14092R1

Targeted detection and quantitation of histone modifications from 1,000 cells

PLOS ONE

Dear Dr. Kelleher,

Thank you for submitting your manuscript to PLOS ONE. After careful consideration, we feel that it has merit but does not fully meet PLOS ONE’s publication criteria as it currently stands. Therefore, we invite you to submit a revised version of the manuscript that addresses the points raised during the review process.

The reviewers agree that some of their previous concerns were addressed. However the reviewers, and I, felt that not all the concerns were adequately addressed. In particular, the concerns about the previous literature and how it is represented in your manuscript needs to be rectified. The biological context discussion of AML also needs to be improved.

We look forward to receiving your revised manuscript.

Kind regards,

C. Michael Greenlief, Ph.D.

Academic Editor

PLOS ONE

Reviewers' comments:

Reviewer's Responses to Questions

**Comments to the Author**

1. If the authors have adequately addressed your comments raised in a previous round of review and you feel that this manuscript is now acceptable for publication, you may indicate that here to bypass the “Comments to the Author” section, enter your conflict of interest statement in the “Confidential to Editor” section, and submit your "Accept" recommendation.

Reviewer #1: (No Response)

Reviewer #2: All comments have been addressed

Reviewer #3: (No Response)

2. Is the manuscript technically sound, and do the data support the conclusions?

Reviewer #1: Partly

Reviewer #2: Yes

Reviewer #3: Partly

3. Has the statistical analysis been performed appropriately and rigorously? 

Reviewer #1: Yes

Reviewer #2: Yes

Reviewer #3: Yes

4. Have the authors made all data underlying the findings in their manuscript fully available?

Reviewer #1: Yes

Reviewer #2: Yes

Reviewer #3: Yes

5. Is the manuscript presented in an intelligible fashion and written in standard English?

Reviewer #1: Yes

Reviewer #2: Yes

Reviewer #3: Yes

6. Review Comments to the Author

Reviewer #1: Authors claimed that "these ions were selected to unambiguously confirm the site of modification". If MS2 spectra are not available, it would be good to provide solid evidence to support these ion transitions.

Reviewer #2: Discussion of multi-omics approaches are added in this version, and I believe it is suitable for publication.

Reviewer #3: In this revised manuscript Abshiru et al. present work that improves the sensitivity on a number of cells analyze basis for proteomic quantitation of histone PTMs. The key step is skipping nuclei isolation for cell numbers less than 1x10^4 and disposing of additional rounds of derivitization. This makes significant advances in sensitive histone proteomic analysis for limited samples in a variety of applications. The overall quality of the work is very good and the sensitivity improvements are substantial. The improvements presented here are significant and important; however, the claims remain somewhat overstated and poorly contextualized. The responsiveness of the authors to some reviewer comments was limited and half-hearted, in other places the response is thorough and appropriate. Many major errors and mistakes in language have been fixed. A major flaw remains in the apparent misreading or possible misrepresentation of the prior literature and the overstatement of the improvements made. I find the response of the authors to the reviewer comments on this subject perplexing bordering on disconcerting. I see no need to misrepresent the value of the current work by creating a straw man when the improvements offered in the current work are real and significant when compared to reality.

Major revision

Crux of the major issue:

You completely misread the Guo paper: The claim of the Guo paper is that the prevailing methods (dating back to ~2009) ARE capable of obtaining results from 50,000 cells NOT that they have an improved method that is capable of such. This evidence is clear and unambiguous.

Comments on prior literature generally:

Certainly, one of the issues is that the prior literature has not been as careful or consistent in presenting quantitative metrics of sensitivity and sample requirements. Sometimes specific sensitivity claims are not even stated while other use differing metrics. This makes proper presentation of the context an effort; however, the authors should make the effort and interpret the literature for the audience. The difference between the “number of cells” and “mass of protein on column” sensitivities is important to this particular effort. The frequent but not universal absence of solid sensitivity claims in prior work should not justify claiming that there are no improvements but rather noting this fact prominently as incomplete information in the literature and that it may be hard to judge. A little reading allows fairly clear inferences about prior work. For example, the Guo paper that is now cited for the clear and comparable “50,000 cells” claim includes this sentence: “Histones were extracted as described previously with minor modification” and cites a Jove paper which is essentially a video version of the 2009 J Proteome Res. One Pot paper. Thus, they are the same procedure with inconsequential modification. It is quite clear from reading the Guo paper carefully that they are not presenting an improved method that is now capable of this. The very title begins with “Assessment of …”. Thus, it is claiming that this is the capacity of the established methods but no one has ever really assessed this capacity properly. “…our study addresses a simple, albeit critical, question about the amount of material required for MS analysis of histone PTMs.” Lots of papers in this area start with large numbers of cells not because it is a requirement but because it is easy, convenient and the work is not sample limited (cell culture or tumors). It is also obvious that skipping the HPLC step as was done in the one pot paper was essential to this sensitivity capacity demonstrated by Guo. This is in fact the primary thesis of the improvements in this work as you state: “reduction in sample processing allows for quantitation of very low input samples.”. Skip nuclei isolation. Skip the second round of propionylation. Even more histones are extracted for low cell counts. There is no doubt that the contribution you are making here is important but do not overstate it. You have made a 5-fold to maybe approaching 50-fold improvement over the established methods that have been in use for over ten years, if only recently assessed for LLOQ by input cell number and not widely used in cell number limited work. Nonetheless, the improvements made here are pretty substantial and enabling of important future work.

AML and biological context:

I agree entirely that care should be applied to not overstating the meaning of the data presented here in understanding the fundamental mechanisms of AML. There is simply not enough statistical power for such claims. However, there are trends that are present and worthy of discussion and comparison to other studies. The concordance with prior work is validating some of which have solid in vivo models, Kaplan-Meier curves and other follow up experiments from the proteomics data. It is also important to discuss the limitations of the data and warn readers not to over interpret it. The concordance of your (limited) results with (limited) prior observation are also validating of the capacity of your improved approach to accurately measure.

References cited:

Separately from misrepresentation of the literature, the authors also seem to have a strong self citation bias and a tendency toward selective exclusion of inconvenient but appropriate literature for comparison.

7. PLOS authors have the option to publish the peer review history of their article (what does this mean?). If published, this will include your full peer review and any attached files.

Reviewer #1: No

Reviewer #2: No

Reviewer #3: No

---

## [Author Response · Author response to Decision Letter 1]

23 Sep 2020

Reviewer #1: Authors claimed that "these ions were selected to unambiguously confirm the site of modification". If MS2 spectra are not available, it would be good to provide solid evidence to support these ion transitions.

Peptide transitions that were used in this manuscript were previously developed and published. Zheng et al. (Zheng Y, Sweet SM, Popovic R, Martinez-Garcia E, Tipton JD, Thomas PM, et al. Total kinetic analysis reveals how combinatorial methylation patterns are established on lysines 27 and 36 of histone H3. Proceedings of the National Academy of Sciences of the United States of America. 2012;109(34):13549-54) initially developed the methods for methylated peptides. One year later, he further exanded this to acetylated peptides (Zheng Y, Thomas PM, Kelleher NL. Measurement of acetylation turnover at distinct lysines in human histones identifies long-lived acetylation sites. Nature communications. 2013;4:2203). 

We have included these references within the methods on pg 7, line 157, as well as a full list of all transitions as a supplemental table (S7 Table).

Reviewer #2: Discussion of multi-omics approaches are added in this version, and I believe it is suitable for publication.

Reviewer #3: In this revised manuscript Abshiru et al. present work that improves the sensitivity on a number of cells analyze basis for proteomic quantitation of histone PTMs. The key step is skipping nuclei isolation for cell numbers less than 1x10^4 and disposing of additional rounds of derivitization. This makes significant advances in sensitive histone proteomic analysis for limited samples in a variety of applications. The overall quality of the work is very good and the sensitivity improvements are substantial. The improvements presented here are significant and important; however, the claims remain somewhat overstated and poorly contextualized. The responsiveness of the authors to some reviewer comments was limited and half-hearted, in other places the response is thorough and appropriate. Many major errors and mistakes in language have been fixed. A major flaw remains in the apparent misreading or possible misrepresentation of the prior literature and the overstatement of the improvements made. I find the response of the authors to the reviewer comments on this subject perplexing bordering on disconcerting. I see no need to misrepresent the value of the current work by creating a straw man when the improvements offered in the current work are real and significant when compared to reality.

We thank the reviewer for their comments on the strong quality of the work submitted. We certainly didn’t intend to offend, misrepresent or overstate the significance of the work submitted and had believed that we had fully addressed these issues in the prior revision. To address these lingering concerns, were have re-written the discussion section to include additional comparisons to the current literature and hope this will now meet the needs of the reviewers and editors of PLoS One.

Major revision

Crux of the major issue:

You completely misread the Guo paper: The claim of the Guo paper is that the prevailing methods (dating back to ~2009) ARE capable of obtaining results from 50,000 cells NOT that they have an improved method that is capable of such. This evidence is clear and unambiguous.

As discussed above, we do not dispute this fact and have re-written the discussion to include a more in-depth comparison with the current histone sample preparation literature (pg. 15, line 336).

Comments on prior literature generally:

Certainly, one of the issues is that the prior literature has not been as careful or consistent in presenting quantitative metrics of sensitivity and sample requirements. Sometimes specific sensitivity claims are not even stated while other use differing metrics. This makes proper presentation of the context an effort; however, the authors should make the effort and interpret the literature for the audience. The difference between the “number of cells” and “mass of protein on column” sensitivities is important to this particular effort. The frequent but not universal absence of solid sensitivity claims in prior work should not justify claiming that there are no improvements but rather noting this fact prominently as incomplete information in the literature and that it may be hard to judge. A little reading allows fairly clear inferences about prior work. For example, the Guo paper that is now cited for the clear and comparable “50,000 cells” claim includes this sentence: “Histones were extracted as described previously with minor modification” and cites a Jove paper which is essentially a video version of the 2009 J Proteome Res. One Pot paper. Thus, they are the same procedure with inconsequential modification. It is quite clear from reading the Guo paper carefully that they are not presenting an improved method that is now capable of this. The very title begins with “Assessment of …”. Thus, it is claiming that this is the capacity of the established methods but no one has ever really assessed this capacity properly. “…our study addresses a simple, albeit critical, question about the amount of material required for MS analysis of histone PTMs.” Lots of papers in this area start with large numbers of cells not because it is a requirement but because it is easy, convenient and the work is not sample limited (cell culture or tumors). It is also obvious that skipping the HPLC step as was done in the one pot paper was essential to this sensitivity capacity demonstrated by Guo. This is in fact the primary thesis of the improvements in this work as you state: “reduction in sample processing allows for quantitation of very low input samples.”. Skip nuclei isolation. Skip the second round of propionylation. Even more histones are extracted for low cell counts. There is no doubt that the contribution you are making here is important but do not overstate it. You have made a 5-fold to maybe approaching 50-fold improvement over the established methods that have been in use for over ten years, if only recently assessed for LLOQ by input cell number and not widely used in cell number limited work. Nonetheless, the improvements made here are pretty substantial and enabling of important future work.

In the discussion, we have attempted to place our work within the context of what was previously done. With regard to the comparison with the Guo paper, we have stated that the results obtained at 50,000 cells when the nuclear isolation is omitted shows no benefit in sensitivity. For 50,000 cells and above, there is no benefit to our revised method and we suggest that the traditional approach may be preferred at these higher cell starting amounts (pg. 16, line 364). 

AML and biological context:

I agree entirely that care should be applied to not overstating the meaning of the data presented here in understanding the fundamental mechanisms of AML. There is simply not enough statistical power for such claims. However, there are trends that are present and worthy of discussion and comparison to other studies. The concordance with prior work is validating some of which have solid in vivo models, Kaplan-Meier curves and other follow up experiments from the proteomics data. It is also important to discuss the limitations of the data and warn readers not to over interpret it. The concordance of your (limited) results with (limited) prior observation are also validating of the capacity of your improved approach to accurately measure.

Given the small “N”, we have been extremely careful not to overstate the findings in the context of AML. In this revision, we have expanded on the role of histone modifying enzymes in blood cancer beginning on pg 16, line 375. We have further shown evidence of the role of H3 K9me3 and SETDB1 levels in AML beginning on pg 17, line 402. 

References cited:

Separately from misrepresentation of the literature, the authors also seem to have a strong self citation bias and a tendency toward selective exclusion of inconvenient but appropriate literature for comparison.

We respectfully disagree that we displayed “strong self-citation bias”. Our research group has been active in histone analysis for over 15 years, publishing over 50 papers in this field, much of it building on past methods developed by our group and others applying it within biological contexts. In the previous revision, just 6 of the 40 total references (15%) were to our prior work. With the additional references added in this revision, this ratio drops to 6/45 total or ~13%.

---

## [Editor Report · Decision Letter 2]

25 Sep 2020

PONE-D-20-14092R2

Targeted detection and quantitation of histone modifications from 1,000 cells

PLOS ONE

Dear Dr. Kelleher,

Thank you for submitting your manuscript to PLOS ONE. After careful consideration, we feel that it has merit but does not fully meet PLOS ONE’s publication criteria as it currently stands. Therefore, we invite you to submit a revised version of the manuscript that addresses the points raised during the review process.

There is a minor point that the authors need to address about Table S7. The manuscript should be accepted after the minor concern mentioned at the end of this letter.

We look forward to receiving your revised manuscript.

Kind regards,

C. Michael Greenlief, Ph.D.

Academic Editor

PLOS ONE

Additional Editor Comments (if provided):

The authors have addressed all the reviewer concerns. The new discussion is well presented.

There is one minor point: On page 25, line 530: There should be a table S7 listed along with its title.

---

## [Author Response · Author response to Decision Letter 2]

30 Sep 2020

We appreciate the positive response to the revised manuscript. We have included the title for table S7 on pg 25, line 526.

---

## [Editor Report · Decision Letter 3]

5 Oct 2020

Targeted detection and quantitation of histone modifications from 1,000 cells

PONE-D-20-14092R3

Dear Dr. Kelleher,

We’re pleased to inform you that your manuscript has been judged scientifically suitable for publication and will be formally accepted for publication once it meets all outstanding technical requirements.

Kind regards,

C. Michael Greenlief, Ph.D.

Academic Editor

PLOS ONE

Additional Editor Comments (optional):

This is an interesting study and worthy of publication. The authors have addressed all of the reviewers' concerns well in the revised manuscript.
---

## [Editor Report · Acceptance letter]

14 Oct 2020

PONE-D-20-14092R3 

Targeted detection and quantitation of histone modifications from 1,000 cells 

Dear Dr. Kelleher:

I'm pleased to inform you that your manuscript has been deemed suitable for publication in PLOS ONE. Congratulations! Your manuscript is now with our production department. 

Kind regards, 

on behalf of

Dr. Charles Michael Greenlief 

Academic Editor

PLOS ONE